# Classroom Digital Teaching and College Students’ Academic Burnout in the Post COVID-19 Era: A Cross-Sectional Study

**DOI:** 10.3390/ijerph192013403

**Published:** 2022-10-17

**Authors:** Wenlong Song, Zihan Wang, Ruiqing Zhang

**Affiliations:** 1School of International Relations, Beijing International Studies University, Beijing 100024, China; 2International Education College, Henan University, Kaifeng 475001, China; 3School of Sociology and Population Studies, Renmin University of China, Beijing 100086, China

**Keywords:** academic burnout, educational psychology, digital teaching and learning, mental health, digital health, post COVID-19 era

## Abstract

The continued development of digital technology and its overuse in teaching and learning in the post-epidemic era have brought about digital health risks, which are associated with academic burnout among college students. This study focused on the relationship between classroom digital teaching and students’ academic burnout and designed the Classroom Burnout Inventory (CBI) and the Classroom Burnout Causes Inventory (CBCI) to conduct a cross-sectional survey of 206 Chinese university students. Correlations and regression analyses were conducted between key factors and burnout values through a path model of “Digital teaching-Teaching & learning process-Causes subjects-Burnout”. The results of the study show that an inappropriate and excessive use of unintegrated digital teaching and learning technologies in the classroom was positively correlated with academic burnout among college students. Burnout levels and the three manifestations were not correlated with students’ gender, grade, and major. In terms of causes, the academic burnout of college students was more correlated with their own personal reasons than with external factors such as teachers, universities, and environments. Integrating digital technology platforms, enhancing teacher leadership in the digital classroom, and strengthening peer support and students’ psychological resilience are all meaningful explorations of academic burnout prevention strategies.

## 1. Introduction

In the post COVID-19 era, the prevalence of mental health problems in Chinese college students is not reassuring; among these mental health problems, academic burnout has become a focus of attention. Chinese university students had to deal with new specific problems such as closed campus management, online study, and employment pressure during the COVID-19 epidemic. Due to their lack of experience in dealing with emergencies and ability to cope with problems, they were more vulnerable to internal trauma and stress reactions such as panic, unease, and anxiety [1]. A youth subculture called “lying flat” has even emerged on the Chinese Internet [2], and college students use “lying flat” to describe their cynical and negative attitude toward learning.

A growing body of evidence now clearly supports the link between pandemic-related stressors and mental health problems [3]. A longitudinal study by Li et al. showed that Chinese undergraduates’ negative mood and anxiety scores increased during the COVID-19 pandemic [4]. Specific data show that the incidence of anxiety and depression among college students during the quarantine management of COVID-19 was of 26.6% and 21.16%, respectively [5]. Researchers have searched for the causes of college students’ psychological disorders from factors such as closure control policies, online learning styles, sleep quality, and physical exercise. Internet addiction, depression, anxiety, academic pressure, and employment pressure are considered to be the main factors affecting college students’ mental health [6]. Long-term online learning also increases the likelihood of academic burnout among undergraduates [7], as “adaptation to online teaching” is an important influencing factor [8]. While the factors of online digital teaching since the epidemic have been seriously neglected, learning is the main task of college students and occupies most of their non-sleeping time, from which the caused psychological stress should be paid attention to and studied.

In the post COVID-19 era, the curriculum for Chinese college students has a tendency to be overly digital. After the epidemic was gradually brought under control, most college students, except those in high-risk areas, were able to return to school, and Chinese universities entered a “new normal” of digital teaching and learning—a combination of online and offline teaching used to ensure that students can attend classes either in the classroom or isolated at home. With multiple learning systems and faculty management systems guiding the entire process, the college students’ entire academic careers are being digitized. This “digital overload” potentially negatively impacts the mental health of college students [9].

However, it is starting to be called into question whether an increasingly deep digitization of education can be justified. How has the deeply digital classroom impacted the mental health of university students? Moderate and appropriate Internet use can be seen as beneficial to learning [10], but an overuse of it might also lead to negative consequences. People who overuse the Internet may not be addicted to the digital world, but rather overuse the Internet as a tool for work or life. The overuse of digital technology can disrupt their academic life and social relationships [11]. Rethinking digital approaches to education can help us better maintain the mental health and academic performance of college students.

The study focuses on undergraduate digital education in China in the post-COVID-19 period. The goal is to investigate whether academic burnout among college students is associated with digital classroom teaching. First, this study may be relevant in identifying the role of digital technology in college students’ learning processes and investigating the real reasons behind their academic burnout. Second, an overview of the manner in which the weariness of university students is impacted by digital education is useful. Third, it offers methods to prevent and treat mental health crises brought on by excessive classroom digital teaching and learning, such as management approaches, instructional tactics, and psychiatric therapy.

To this end, this paper proposes the following hypotheses and attempts to test them through cross-sectional data obtained from an online survey of college students.

**H1.** *Academic burnout among college students is not significantly correlated with students’ gender and subject background*.

**H2.** *Academic burnout among college students is mainly influenced by external factors such as digital techs, teachers, schools, and environments*.

**H3.** *Inappropriate use of digital teaching technologies shows a positive predictive relationship with college students’ academic burnout*.

## 2. Literature Review

### 2.1. Classroom Digital Teaching

Digital teaching is the practice of teachers and students engaging in instructional activities inside a digital learning environment while adhering to contemporary educational theories and guidelines, employing digital teaching resources, and utilizing digital teaching techniques to develop talents. As a different educational model from traditional teaching, it pushes for a closer integration of the two, not just the use of digital technology for the educational process [12]. This necessitates the provision of pedagogical, technological, and content capabilities [13], the development of an effective teaching and learning environment [14], and the availability of interactive computer programs and distance learning [15]. At a micro level, digital teaching means that educational technology, teacher–student interaction, and classroom pedagogies have shifted from the industrial age to the digital age [16]; at a macro level, it is about optimizing the operations, strategic direction, and value proposition of higher education institutions to form an education system that is compatible with the digital age.

Research in this area is conducted from three main perspectives: technological, social, and institutional. The first focus of the technological perspective is how digital technologies have altered teaching and learning paradigms. In comparison to traditional teaching and learning, digital teaching is more flexible and effective [17], and learners have access to a wider variety of instructional content. Face-to-face instruction combined with digital components can aid in creating a high-quality, student-centered teaching and learning paradigm in the post-epidemic age [18]. Second, the social perspective relates to the relationship between society and higher education, as well as collaboration on a global scale. According to one viewpoint, the epidemic has increased the pressure that is required for digital teaching and learning and has aided in the forward-thinking transmission of knowledge [19]. Third, the institutional perspective explores the core elements of teaching and learning, namely students, curricula, teachers, professions, and universities [20]. Students, teachers, and university staff all face fundamental change processes and a need to improve their own digital teaching competencies as a result of the challenges posed by digital teaching and learning [21,22,23]. Universities with high levels of autonomy and decentralized organizations are better able to create the institutional and structural conditions for innovation in digital teaching and learning [24].

Some academics, however, contend that there is a risk that new media will simply be incorporated into conventional teaching methods [25], that digital technologies will be ineffectively incorporated into the teaching and learning process, and that the teaching and learning process will essentially remain a teacher-centered activity [26]. Others have questioned the value of digital teaching and learning, claiming that technology falls short of fully replacing traditional face-to-face instruction and offers only a modest contribution to raising student achievement [27,28].

In the post-epidemic classroom, digital teaching and learning is seen as more of a blended approach. Here, digital education is employed in conjunction with conventional techniques [29]. Despite being widely acknowledged by students, it had no appreciable impact on final exam scores when compared to conventional instruction [30]. Additionally, a number of studies have revealed that in the post-epidemic era of digital education, psychological barriers prohibit many students from actively participating in classroom interactions, and professors find it challenging to engage students and build two-way communication [31].

Digital education in China did not start during the COVID-19 epidemic, but 20 years earlier, when China had had access to the global Internet for less than a decade. Driven by digital multimedia and Internet teaching technologies, the traditional university classroom has been profoundly transformed, with various electronic devices becoming tools for teacher–student interaction This computer application-based multimedia teaching model has made an impact on the traditional blackboard writing.

Digital education in Chinese universities has gone through three stages. The first period (2000–2010) was “e-Learning”, which focused on connecting educational resources to the Internet, introducing multimedia technology into traditional classrooms, and gradually establishing a platform for sharing educational resources such as online catechisms. The “digital campus” was built in terms of network construction, digital learning terminals, learning support systems and information security, and a digital environment that integrates various teaching-learning systems [32]. Teachers in higher education institutions are equipped with desktop computers, laptops, and tablets. In the Chinese Project 211 institutions, 95.83% of faculty members were equipped with desktop computers and 91.67% with laptops. Before the epidemic, more than 80% of universities built online learning platforms [33]. The use of multimedia images and videos reduces the overwhelming nature of text and helps students manage cognitive load, thus reducing the difficulty of memorization. A controlled pilot study showed that students’ use of laptops in the classroom had little effect on teaching strategies, but significantly enhanced learning [34].

The second period (2011–2019) is mobile learning (m-Learning), which establishes intelligent teaching tools that integrate PowerPoint (PPT) slides, Massive Open Online Courses (MOOCs), and mobile devices. The deep integration of MOOCs, live classes, and offline face-to-face classes exposes college students to a deeply digital educational environment [35]. This type of immersive digital classroom occurs when students use the Internet more frequently for learning, such as writing assignments using a word processor, creating presentations, and communicating with classmates and teachers via email.

The 2020 COVID-19 pandemic accelerated the digitization of teaching and learning, driving digital learning in China into a third phase—”smart learning” (s-Learning). With China’s Ministry of Education’s Education Informatization 2.0 Action Plan, s-Learning, marked by the “Smart Classroom”, was launched. The “smart classroom”, also known as the “classroom of the future”, is a learning environment that integrates big data, Internet of Things, artificial intelligence, and other technologies on the basis of traditional classrooms, with deep teaching interaction as its core. It was previously developed and implemented in other countries, which promoted the transformation from a digital learning environment to a smart learning environment [36]. A large part of the current learning environment for college students was dominated by digital technology, including technology-mediated learning [37]. During this period, a digital media hybrid teaching model based on mobile Internet technology was explored, which consists of four parts: front-end analysis, online independent learning, offline teaching, and online evaluation after class [38]. According to the China Internet Network Information Center’s (CNNIC) data, driven by “home learning” during the epidemic, the scale of Chinese online education users increased from 232 million before the epidemic in June 2019 to 423 million in March 2020 (an increase of 82%), and the proportion of Chinese Internet users increased from 27.2% to 46.8%. As the epidemic is effectively controlled and universities resume normal teaching, the size of China’s online education user base declines since June 2020 to 325 million in June 2021, but is still much higher than the pre-epidemic level due to the adoption of a blended “online + offline” teaching approach [39].

### 2.2. Academic Burnout

Academic burnout, primarily involving the student population, is a category of career burnout. It is often interpreted as a learning-related syndrome characterized by maladaptive emotional and physiological responses to a long-term exposure to stressful events [40,41].

Academic burnout has three main manifestations, namely exhaustion due to the demands of learning (emotional exhaustion), a cynical or transcendental attitude toward one’s own learning (depersonalization), and a sense of incompetence and difficulty in generating satisfaction as a student [42] as well as a negative evaluation of the educational environment (low personal achievement) [43]. The main characteristic of burnout is a combination of exhaustion (low activation) and cynicism (low identity), while the opposite of academic dedication is characterized by vitality (high activation) and dedication (high identity) [44].

How does academic burnout affect students? On the one hand, it can lead to a lack of motivation in university students. The relationship between the use of digital technology and the Internet and college students’ motivation and academic performance has been of interest to researchers. The self-determination theory of motivation and engagement suggests that students determine their own learning behaviors based on external resources [45], and that students become less motivated when their psychological needs (competence, autonomy, and relatedness) are not met [46]. When traditional non-digital learning methods fail to meet their psychological needs, academically burned-out college students accumulate dissatisfaction and may seek the comfort of the Internet and become more addicted to it. Concerns about the impact of digital technology on academic performance are particularly acute when college students have a “problematic internet use” (PIU) or are addicted to the Internet [47]. Problematic Internet use can create feelings of isolation, which directly affects motivation to use learning strategies and can lead to other chronic psychological disorders [48].

On the other hand, academic burnout is related to the academic performance of university students. Some earlier studies showed no significant association between student burnout and academic performance [49], but more recent cross-sectional studies have designed and hypothesized that student burnout is an influential factor in academic performance [50,51]. For example, a study of nursing students showed a positive association between academic network abuse and academic burnout (r = 0.305, *p* < 0.001), but a negative association between Internet abuse and academic performance (r = −0.478, *p* < 0.001) [52]. The overuse of the Internet or a social network addiction has a negative and significant impact on students’ academic performance [53,54], and a meta-analysis based on 100,000 students showed that burnout leads to poorer academic performance [55].

### 2.3. Classroom Digital Teaching and Academic Burnout

College students are a susceptible population for Internet addiction (IA). The particular period affected by COVID-19 may increase college students’ online social media use, and the risk of IA is further exacerbated as anxiety about the epidemic deepens [56]. In a meta-analysis study, the prevalence of Internet addiction disorders among Chinese college students before the COVID-19 outbreak was of 11.3% [57]. Additionally, after the COVID-19 outbreak, this number was reported to have risen to 28.4% [58]. Imani et al. and Salmela-Aro et al. reported a positive correlation between Internet addiction and burnout [59,60]. To what extent does college students’ screen time affect their mental health? This question has generated disagreement in a wide range of studies [61]. Wästlund et al. concluded that although young people experience lower psychological well-being on the Internet, no significant correlation between Internet use and psychological health was found [62]. LaRose et al. investigated and found a low correlation between Internet use and depression (r = −0.02) [63]. A study of adolescent Internet users in the UK found that moderate digital engagement was not associated with well-being, but very high levels of use may have a small negative association [64].

How does the classroom digital teaching model affect university students’ academic burnout? There are certain peculiarities in the academic burnout of Chinese university students in the post COVID-19 era.

First, “Smartphone dependence” during study inhibits college students’ motivation to learn and makes them indifferent to and detached from learning [65]. The current generation of young people, sometimes called “digital natives”, have built their social identities in the real and cyber worlds constituted by digital devices and the Internet since childhood. In contrast to digital immigrants, digital natives use cell phones as an essential “e-organ”, are accustomed to receiving information rapidly, prefer multitasking and random access, prefer instant feedback and reinforcement, and prefer visual to auditory or textual information. They lack patience for monotonous classroom lectures and are unable to focus on learning for long periods of time, so they often use their phones in class. Digital tools can distract them from learning and may cause a lower motivation to learn [66]. Academic burnout has been linked to both social and smartphone use, and Walburg et al. found that academic burnout was associated with a higher frequency of cell phone social app use [67]. Gundogan confirmed that academic burnout was associated with problematic cell phone use, especially during the COVID-19 pandemic [68]. Problematic cell phone use can worsen into “cell phone dependence”, wherein one feels restless and unconscious when away from one’s phone for too long [69] and unable to study and live normally. He and Wan et al. compared the performance of cell-phone-dependent and non-cell-phone-dependent adolescents in terms of life events, academic burnout, and mental health. The data showed that there were significant differences between the two and that cell-phone-dependent individuals scored significantly higher on each variable than non-cell-phone-dependent individuals [70].

Second, teachers’ inappropriate use of digital instructional technology hinders emotional interaction and affects students’ engagement in learning, psychological satisfaction, and personal fulfillment. Digital technology as a “teaching aid” refers to the use of multimedia to enhance teacher-led instruction. Despite high expectations, the fragmented way in which technology has been introduced into the educational process in most schools has been a barrier to the effective use of technology for teaching and learning, and as a result, early models of digital aids have not had a significant impact on improving academic achievement [71]. Teachers’ pedagogy, teaching effectiveness, quality of PPTs and ability to control the classroom were the most important factors affecting the quality of education [72,73]. Teaching style, personal computer use, and technology-related training all played a role in how much technology was used in the classroom and how it was used [74]. The vast majority of teachers are only at the level of using digital technology to provide a convenient means of teaching and to make it more effective, often in a superficial way. An overly active classroom atmosphere distracts students, especially when there is no pre-reading or notes for instruction, resulting in students not being able to accurately remember 80% of the material covered in class [75]. Conversely, an overly dull classroom results in an inadequate interpersonal atmosphere among students, who then cannot effectively engage in learning. Some teachers rely too much on PPTs, often just reading aloud pages of text-filled presentations in sequence, neglecting to interact with students, which inhibits students’ enthusiasm for class participation. The only thing students can do is to look at the screen, take notes, and be submerged in a large amount of material, which makes learning both boring and fatiguing [76]. The lack of direct contact between students and instructors, especially in the online class model, can be a significant stressor for academic burnout [77].

Third, the large class sizes of digital classrooms make it unlikely to meet the participation needs of every student, which is associated with inappropriate student behavior in the classroom. With the rapid expansion of Chinese universities, university classroom sizes are often larger than they should be. Additionally, the larger the class, the more difficult it is for teachers to organize effective classroom activities; this leads to less teacher–student interaction and less interpersonal interaction among students, making the classroom more exhausting—one of the causes of academic burnout. Large classes lead to students with different learning abilities feeling neglected and triggers depersonalization. In a large class of 60 or more students, teachers are unable to keep the differences in students’ levels within their control and are forced to follow pre-designed lesson plans. On the one hand, when learning stress is beyond their capacity, avoidance coping strategies play an important role in burnout. Students who are not computer savvy may have to spend more time improving their web-based information retrieval skills rather than focusing on course material [78]. Students who are poor learners simply miss classes because they cannot keep up with the course. On the other hand, when courses struggle to meet the students’ needs, students lose motivation to participate in class. The more advanced students find the pace too slow and spend class time on other things, such as the Internet. The inappropriate use of digital technology can reduce students’ perception of the school climate, and this negative state can reduce the sense of belonging and identification with the university [79]. Thus, students’ perception of the classroom environment constitutes an important variable in predicting their level of academic burnout [80].

Fourth, in the post-epidemic period, multiple digital teaching and learning platforms have left college students exhausted by feeling “falsely busy”. Digital teaching and learning technologies in non-school settings have changed the “learning styles, strengths, and preferences” of college students, reshaping the way they access information, communicate, and learn [81]. The “online + offline” blended teaching and learning model in the post COVID-19 period, which overlaps multiple disconnected learning platforms, inevitably continues to distract students, consume their time, and challenge their ability to multitask and coordinate resources. This can lead to students feeling like they are always rushed, resulting in a sense of exhaustion [82], yet not accomplishing learning objectives well. There is such a vast amount of knowledge and information covered in e-Learning that students cannot rely on 40 min to digest it in class, which leads to brain fatigue and an inability to focus on the lecture. Most students follow the content arranged in the courseware step-by-step and fail to catch the main points. This is the reason why electronic courseware is very informative but does not produce the corresponding learning efficiency.

## 3. Methods

### 3.1. Participants

This study used a quantitative research method to collect data by distributing an online questionnaire to Chinese college students who were experiencing digital teaching in the post-epidemic period. We distributed the questionnaire through Tencent Questionnaire (https://wj.qq.com), a professional survey platform in China. The online survey was available from 4 August 2022 to 23 August 2022, and eligible college students were invited to participate in the online survey through snowball sampling. Informed consent was obtained from participants prior to the survey, all students were volunteers, and no one received any compensation or credit for participation. The questionnaire comprised 3 consecutive pages, and the platform prompted respondents to answer each question completely before proceeding to the next page, and only after all questions were completed, they could be submitted. Therefore, we ended up with all valid questionnaires returned and no questionnaires with missing data. The questionnaire platform also collected information such as the province of the respondent and the time spent filling out the questionnaire. The students who participated in the official survey were 206 college students from 23 provinces, municipalities, and autonomous regions across China.

### 3.2. Measures

This questionnaire is divided into three parts. The first part collects basic information about the students, including their gender, the number of years they have been in college, their high school background, and the type of college major. The second part is the “Classroom Burnout Inventory” (CBI), which is designed to test the level of academic burnout of college students. We combined the Maslach burnout inventory–student survey (MBI-SS) [83] and the “Adolescent Student Burnout Inventory” developed by Wu et al. [84] as well as adapting some of the questions to meet the needs of this study, which involved classroom digital teaching. The questions on this scale are more specific than the traditional academic burnout inventory, focusing on classroom learning from the whole learning process, for example, by replacing “I cannot get a sense of accomplishment in my learning” with “I cannot get a sense of accomplishment in my classroom learning”. The Classroom Burnout Inventory has 16 questions and uses a five-point Likert scale, with 1 being strongly disagree and 5 being strongly agree. Three subscales were used to measure three dimensions of classroom burnout, namely emotional exhaustion (6, 9, 13, 15, 18, and 20), inappropriate behavior (5, 8, 11, 14, and 16), and low achievement (7, 10, 12, 17, and 19), and scoring was done from a 1–5 scale to indicate the degree of academic burnout compliance, and scores for each dimension were calculated by cumulative summation. There were 8 reverse questions (5, 7, 8, 10, 11, 12, 17, and 19), which were recoded after the questionnaire was returned. The CBI scores ranged from 16 to 80, with higher total scores obtained indicating greater academic burnout.

The third part of the questionnaire is the Classroom Burnout Causes Inventory (CBCI), which aims to find the causes of academic burnout and classroom performance among college students. We designed 17 questions on a five-point Likert scale, with 1 being strongly disagree and 5 being strongly agree. Three of the subscales investigate the relationship between students’ own causes (21, 27, 29, 32, and 33), teachers’ and schools’ causes (22, 25, 26, 28, 31, 35, and 36), and environmental causes (23, 24, 30, 34, and 37) and academic burnout in the classroom. In order to determine the validity and appropriateness of the questionnaire scale’s questions, regarding the validity of the questionnaire design, we tested them on the SPSSAU analysis platform (https://spssau.com/) using item analysis. The principle is to first sum the analyzed items and then divide them into high and low groups (bounded by 27% and 73% quartiles, respectively), and then use the *t*-test to compare the differences between the high and low groups; if there are differences, the inventory items are appropriately designed, and if not, the inventory items cannot distinguish the information and the unreasonably designed questions should be removed. The analysis revealed that all 17 items of the causes for the classroom burnout scale showed significance (*p* < 0.05), implying that all 17 items in total were well differentiated.

### 3.3. Data Processing

After recoding, regarding the reverse questions, Cronbach’s reliability analysis was performed on all questions of the questionnaire using R software. Two of the questions, 19 and 20, had correction item total correlation (CITC) values significantly below 0.4 and were excluded. The reliability analysis revealed that the reliability coefficient value α was 0.949, which was greater than 0.9, thus indicating a high quality of reliability of the study data [85].

The validity of the design rationality of the attitude inventory question data was tested using factor analysis to detect whether the research items were reasonable and meaningful. A comprehensive analysis was performed by KMO values, commonality, variance explained values, and factor loading coefficient values, in order to verify the validity level of the data. It was found that the commonality values of all the research items were higher than 0.4, indicating that the information of the research items could be extracted effectively. In addition, the KMO value is 0.909, which is greater than 0.8, and the study data are very suitable for extracting information (a side effect of good validity). The explained variance values of the four factors are 21.069%, 14.289%, 11.669%, and 4.756%, respectively, and the cumulative explained variance after rotation is 64.094% > 50%. This means that the information content of the study term can be extracted effectively [86] (see Figure 1).

We performed factor analysis on the data to downscale the original data. KMO = 0.91 and the *p*-value of Bartlett’s test was much less than 0.05, so these data set are very suitable for factor analysis. According to the results of parallel analysis of the gravel plot, four common factors need to be extracted. The questionnaire investigates the relationship between college students’ classroom performance and digital technology use, and the two are correlated. Therefore, the oblique rotation method was used, and the results are shown in Figure 2.

From the analysis results, the questions in the questionnaire can be divided into 4 major groups: the first major group of questions includes questions 22, 23, 24, 25, 26, 28, 30, 31, 33, 34, and 35; the second major group of questions includes questions 5, 7, 8, 10, 11, 12, and 17; the third major group of questions includes 27, 29, 32, 36, and 37; the fourth major group of questions includes questions 6, 9, 13, 14, 15, 16, 18, and 21. From the content of the questions, we can find that the first group of questions focuses on students’ satisfaction with the classroom teaching format, which can be summarized as “Ability of teaching”; the second group of questions focuses on students’ acceptance and subjective performance of the classroom teaching content, which can be summarized as “Learning receptivity”; the third group of questions focuses on the degree of attractiveness of the classroom to students, which can be summarized as “Appeal of teaching”; the fourth group of questions focused on students’ enthusiasm for learning in the classroom and their ability to actively explore, which can be summarized as “Learning motivation”. In summary, the results of the data analysis reveal that the current state of learning is influenced by two main subjects and four dimensions: classroom teaching (ability and appeal) and student learning (receptivity and motivation).

In order to avoid inconsistencies in the total scores due to the different number of subscale questions, we intervalized the total score of the “low personal achievement” subscale of the classroom burnout scale (5–25) and kept the total scores of the other two subscales the same in the analysis of the relationship between the total scores of the subscales and other variables. Similarly, the total score of the ”teacher & school causes” subscale was intervalized (5–25).

### 3.4. Data Analysis

#### 3.4.1. Descriptive Statistics

We used the SPSSAU (Version 22.0) tool (Beijing, China) for analysis. A total of 206 people were surveyed in this study, of which 62 were male and 144 were female, which corresponds to 30.1% of male participants and 69.9% of female participants. From the geographical distribution of the research platform statistics, 54% of the respondents were from Beijing, and the rest were distributed in 22 of the 36 Chinese provinces. The study shows that students in different grades have different levels of mastery of study methods and familiarity with their majors, which may affect their academic attitudes. The online teaching model for college students during the epidemic closure period posed some new problems for different groups, for example, online education may hinder the understanding of professional knowledge of first-year students [87]. In terms of the duration of subjects’ enrollment: from 0 to 1 year, 29 respondents (14.1%); 2 years, 44 respondents (21.4%); 3 years, 54 respondents (26.2%); 4 years, 38 respondents (18.4%); and 4 years and above, 41 respondents (19.9%) (see Table 1).

In most provinces in China, high school is divided into arts and science, with some provinces making no distinction. The learning background of high school determines how difficult it is for students to major in college. For example, if a student with a science and technology high school background enters university to study arts and history, then he may find it challenging due to the lack of knowledge base and professional foundation, which affects his academic performance and academic burnout level. In this survey, the number of students with high school learning backgrounds favoring liberal arts totaled 117 (57.8%); favoring science totaled 66 (32.0%); and neither liberal arts nor science totaled 23 (11.2%).

There are differences in the applicability of digital education in different majors, for example, for some science and technology majors that focus on experimental operations, online learning may not be as effective as expected, and college students may experience frustration due to the difficulty in mastering practical skills. The categories of university majors in this questionnaire are mainly liberal arts, with 176 people (85.4%), and science and technology, with 30 people (14.6%).

The Classroom Burnout Inventory counted the burnout values of college students with a mean score of 42.413 (16–80), of which emotional exhaustion had a mean score of 14.024 (5–25), misbehaves had a mean score of 13.010 (5–25), and low personal achievement had a mean score of 13.665 (5–25). Additionally, the mean scores of each subscale in the classroom burnout causes scale were as follows: personal causes (13.825), teacher and school causes (13.658), and environmental causes (12.850). The higher emotional exhaustion scores reflect the greater physical and mental impact of digital teaching on college students, and the long hours of screen study leave college students with insufficient energy for physical activity, which indirectly affects sleep and physical health. Compared to finding external causes, college students more often blame themselves for the psychological disorders triggered during classes, such as lack of appropriate digital learning methods, lack of scientific course study planning, and insufficient emotional regulation ability (see Table 2).

#### 3.4.2. Correlation and Regression Analysis

Correlation analysis was used to examine the correlations between gender, grade level, high school academic background, college major category, and class burnout. We used Pearson correlation coefficients to indicate the strength of the correlations. In terms of gender, 12 of the 14 items do not show significance, and the correlation coefficient values are close to zero, indicating that the level of classroom burnout has little relationship with gender (only questions 9 and 13 show that male students have more difficulty maintaining enthusiasm in class). The correlation coefficient values are all close to 0 for the years of enrollment, high school study background, and college major, indicating that these three items are not related to the level of classroom burnout. None of them show significant differences (*p* > 0.05) when analyzing the relationship between gender and three levels of academic burnout: low mood, inappropriate behavior, and low achievement, by *t*-test [88]. Previous research studies have also found that there is no significant difference between students’ burnout levels and gender [89]. The *t*-test and Pearson correlation analysis (Pearson correlation coefficient near 0) also demonstrated that there were no significant differences in burnout values between students’ major categories (arts or science).

Using personal causes, teacher and school causes, and environmental causes as independent variables and the total value of classroom burnout as a dependent variable for linear regression analysis, it can be seen from Table 3 that the model Equation is:Total value of classroom burnout = 18.264 + 0.995 × Personal causes + 0.391 × Teacher and school causes + 0.393 × Environmental causes

The model’s R-squared value is 0.614, which means that personal causes, teacher and school causes, and environmental causes can explain 61.4% of the variation in the total value of classroom burnout. When the F-test was conducted on the model, it was found that the model passed the F-test (F = 107.019, *p* = 0.000 < 0.05), which means that at least one of the personal causes, teacher and school causes, and environmental causes have an effect on the total value of classroom burnout. In addition, the model was tested for multicollinearity and found that all VIF values were less than 5, which means that there is no covariance; and the D-W values were around the number 2, which means that the model is not autocorrelated and that there is no correlation between the sample data and the model is good. The final analysis showed that the regression coefficient value of personal causes was 0.995 (t = 6.308, *p* = 0.000 < 0.01) and the regression coefficient value of teacher and school causes was 0.995 (t = 6.308, *p* = 0.000 < 0.01).

The regression coefficient value for teacher and school causes was 0.391 (t = 2.077, *p* = 0.039 < 0.05), and the regression coefficient value for environmental causes was 0.393 (t = 2.351, *p* = 0.020 < 0.05), concluding that personal causes, teacher and school causes, and environmental causes all have a significant positive effect on the total value of classroom burnout [90].

The results above can be visualized by scatter plotting the data. The scatter data linear fit equations are as follows. academic burnout total = 20.683 + 1.572 × Personal causes, with an R-squared value of 0.575. Academic burnout total = 20.675 + 1.592 × Teacher and school causes, with an R-squared value of 0.492. Academic burnout total = 23.612 + 1.463 × Environmental causes, with an R-squared value of 0.479. From Figure 3, it can be seen that personal causes contribute to a higher value of burnout compared to environmental factors, with a greater degree of dispersion.

Students’ academic burnout is associated with well-being related to school activities (e.g., learning, attending classes) [91]. Specifically, individual, teacher–school, and environmental factors have different levels of influence on the three different manifestations of academic burnout in the classroom. As can be seen in Table 4, the coefficient *p* < 0.01 between all variables reported a high correlation. Among them, classroom misbehavior triggered by personal reasons was particularly prominent (correlation coefficient 0.719). Burnout due to school and environmental factors was more pronounced in students’ mental and emotional well-being (correlation coefficient 0.653), leading to physical and mental exhaustion.

In order to know more precisely the effect of classroom digital teaching methods on students’ burnout, we screened out the questions involving Internet and digital factors (23, 28, 31, 32, and 36) from the Classroom Burnout Causes Inventory questionnaire and conducted correlation analysis with burnout. The correlations between emotional exhaustion, misbehaves, and low personal achievement and the five questions were studied, and the Pearson correlation coefficient was used to indicate the strength of the correlations. Although most of the knowledge can be obtained through an Internet search, most of the college students believe that the teaching process is important for the absorption of knowledge, but the learning atmosphere of the classroom lacks engagement and interaction. The digital teaching model is also not conducive to fostering an atmosphere of peer learning, which sometimes makes it difficult to raise enthusiasm for learning. Teachers inappropriately use digital teaching technology or just use it as an aid, mechanically reading from PowerPoint slides, which becomes an even less stimulating environment than a traditional classroom.

Question 28 on the Causes of Burnout Inventory reported the highest mean score (3.3) of all questions on the scale, with 72.4% of respondents agreeing that “some teachers are uninterested in the classroom due to over-reliance on PowerPoint presentations,” which also had a stronger association with students’ emotional exhaustion (correlation coefficient 0.449). The problem of inappropriate use of cell phones in the classroom also plagued college students, and the results of the study show its correlation with three manifestations of academic burnout, with the highest correlation for inappropriate behavior (correlation coefficient 0.585). This also reveals an ambivalence: most college students can neither tolerate a study space without the Internet nor resist the temptation to be distracted by browsing smartphones, websites, and social software during class. The mean score for question 36 was 3.1, with 67% of college students believing that online self-study on the Web would reduce their motivation to learn and lead to burnout-related misbehavior (see Figure 4).

#### 3.4.3. Path Analysis

How does classroom digital teaching affect students’ psychological state? We conducted a model regression analysis by exploring the pathways among digital teaching, teaching and learning’s four factors, burnout’s three causes, and classroom burnout. The standardized path coefficient values of digital teaching factors for the four factors of “teaching-learning” were all greater than 0 (ability to teach, 0.896; appeal of teaching, 0.683; learning receptivity, 0.453; learning motivation, 0.693), and their paths all showed a significance at the 0.01 level (*p* = 0.01). There was a significant covariance (correlation) between learning motivation and personal causes. There is a significant positive covariance relationship between the standardized path coefficient of 2.570 > 0, and this path shows a significance at the 0.01 level (z = 3.506, *p* = 0.000 < 0.01). There was a significant positive covariate relationship between appeal of teaching and environmental causes. Teacher and school causes did not show a significant relationship with environmental causes (z = 1.680, *p* = 0.093 > 0.05), thus indicating that there was no significant relationship between teacher and school causes and environmental causes. Personal causes and teacher and school causes had a significant positive effect on academic burnout total. The standardized path coefficient values were 0.700 > 0 and 0.086 > 0, respectively, and their paths showed significance at the 0.01 level. Additionally, environmental causes showed a significant negative effect relationship on academic burnout total. The standardized path coefficient value is −0.216 < 0, *p* = 0.000 < 0.01. Its correlation and path are shown in Figure 5. By conducting linear regression analysis with direct digital teaching factors as the independent variable and academic burnout total as the dependent variable, we found that digital teaching factors have a regression coefficient value of 1.528 (t = 12.759, *p* = 0.000 < 0.01), implying that digital teaching factors have a significant positive relationship on academic burnout total, and the model R-squared value is 0.444, implying that digital teaching factors can explain 44.4% of the reason for the change in academic burnout total. It passed the F-test (F = 162.803, *p* < 0.01), which also means that the model construction is meaningful.

This section may be divided by subheadings. It provides a concise and precise description of the experimental results, their interpretation, as well as the experimental conclusions that can be drawn.

## 4. Discussion

The present study points to a positive correlation between the increasing digital model of teaching and learning and academic burnout among Chinese university students. Previous studies have repeatedly indicated that college students with burnout symptoms may have difficulty controlling their emotional responses and may also be at increased risk of more serious psychological disorders [92], such as anxiety, depression, and PTSD (post-traumatic stress disorder) symptoms [93]. Educators’ over-reliance on technology and neglect of college students’ psychological conditions are worthy of our caution. In the process of reflecting on the psychological side effects of digital technology on college students, solutions to this problem need to be explored.

### 4.1. Side Effects of Excessive Digital Teaching and Learning: Technology Dependency and Classroom Burnout

It is clear that digital technologies are not being used appropriately and lack efficient integration in the university classroom. The continuous emergence of digital instructional technology has been installed as a new digital tool module in the learning process of university students. Chinese college students in the post-COVID-19 era may use 3–5 types of video conference software and 1–3 online study management platforms at the same time, which make them feel busy and is detrimental to their academic performance and mental health. The plethora of digital learning tools causes distress to students (Q31 and Q36). Students’ burnout in the classroom originates not only from themselves, but also from teachers, school management systems, and the macro environment.

The rapid advancement of digital education reform is also a challenge for teachers and has led to higher levels of burnout. Compared to traditional teaching, the post-epidemic “online + offline” hybrid teaching requires more effort in curriculum design than before the epidemic, and teachers are under more pressure to prepare lessons. According to statistics, more than 90% of teachers believe that online teaching requires more time and effort; more than half of the teachers said that the preparation time is one to two times longer than usual. Especially for teachers over 45 years old [94], as “digital immigrants”, the very task of mastering digital teaching and learning technologies is a great challenge for them.

In terms of educational resources, some of the paper-based textbooks and reference books are lagging behind in content updates. The information in some textbooks contradicts the latest academic resources on the Internet, and some of the theories in them have even been eliminated by the academic community, which can also hinder students. The development and operation of online courses such as MOOCs in Chinese universities face challenges such as insufficient technical support, insufficient training in course design, and lack of national standards for course design and development as well as platform standards [95].

In terms of classroom management mechanisms, the school has inadequate supervision tools of both teachers and students. From the monitoring screen of the central control center, most of the teachers’ classroom digital teaching is mainly showing PowerPoint slides, followed by playing videos, and there are a few courses mainly using audio recording (such as foreign language majors) and physical projection. Some teachers’ multimedia courseware is only filled with large text, partly due to the teachers’ insufficient skills, and much more because the school only pays attention to whether the teachers use multimedia technology, without assessing the classroom interaction and quality of digital teaching [96]. On the other hand, due to the lack of guidance, there is also a great deal of fragmentation in the informal learning of college students in the virtual world [97], or addiction to social media, entertainment, or online games.

### 4.2. How to Develop Appropriate Digital Education in Post COVID-19 Universities

The main view of the contemporary learning theory (e.g., social shared cognition, contextual learning, activity theory, ecopsychology, and case-based reasoning) suggests that learning is a process of active meaning construction, social collaborative exchange, and daily practical participation [98]. In this process, students complete meaning-making, socialization processes, and learn community building. The face-to-face conversation approach, derived from Socrates and Confucius, has proven effective in practice for thousands of years. A fully digital model of education does not replace the traditional model of teacher–student spiritual interaction and emotional replenishment of peer learning. There is a supply and demand gap between students’ out-of-school digital learning preferences and needs and schooling that may lead to reduced engagement in classroom learning [99]. The era of the epidemic will eventually pass, and it is worthwhile for researchers to explore how to find a balance between new technologies and traditional teaching models, and thus explore a digital education model that is beneficial to students’ mental health in the post COVID-19 era.

First, universities should integrate and streamline digital education tools. The integration of digital teaching tools with different functions through the interconnection and integration of technology platforms makes it easier for students to operate and for schools to manage. In February 2022, China released “Highlights of the Ministry of Education’s Work in 2022”, which proposes to implement a strategic initiative on education digitization, explore the construction of smart classrooms and smart courses in universities, deepen the application of online learning spaces, and improve classroom teaching models [100]. If implemented properly, a number of technology-mediated programs can support student development [101]. Beyond hardware devices and software systems, it is more important to develop work plans that take into account the details of the educational environment [102]. The “smart learning” scenario also needs to be relevant considering the informal learning needs of students who use computers more than mobile devices to handle learning tasks [103].

Second, teachers’ leadership in the digital classroom should be improved. Our survey showed that students were not highly satisfied with the teachers’ leadership in the digital classroom (Q22, Q26, Q28). Teacher leadership has a role in increasing college students’ academic motivation and performance and reducing levels of academic burnout, and transformative teacher behaviors encourage students to engage in classroom learning and interactive processes [104]. Teacher training in digital skills needs to be enhanced, the inappropriate use of digital tools needs to be regulated, and teachers need to be encouraged to explore and enhance classroom leadership. Teacher behaviors that seek to motivate students will result in higher levels of functioning, increased interest and engagement in the class, and elevated critical thinking among college students [105]. Establishing common norms in the digital classroom can help discipline student misbehavior, as students who deviate from norms appear to be more likely to be denied access or interaction by their peers [106]. On the other hand, teacher emotional support can have a mitigating effect on students’ academic burnout [107]. Teachers can design a moderate digital classroom environment to meet students’ needs for knowledge acquisition, emotional connection, personal fulfillment, and collective belonging. To prevent information overload and lack of emotional interaction caused by excessive digitization, teachers should also adjust their digital teaching workload and teaching strategies in a timely manner to help students adapt to the pace of instruction and manage their needs [108].

Last but not least, students should be encouraged to explore strategies to prevent and resist academic burnout. Prevention strategies and remedies are necessary to avoid academic burnout as a result of digital technology abuse or Internet addiction among college students. Our questionnaire reveals that students explore self-based solutions to burnout by developing study plans and self-regulation (Q5 and Q29). Adolescents need to be taught digital technology literacy and enhance their social and emotional learning as well as self-regulation strategies [109]. Students can escape burnout by increasing their psychological resilience and seeking external support. When academic burnout occurs, resilience can help the person get rid of negative symptoms in time to pull that person back on track [110]. A better understanding of the COVID-19 disease and an active lifestyle can help reduce the risk of Internet addiction and academic burnout [111]. This will allow students to find the right way to de-stress instead of being addicted to their phone screens. When faced with academic challenges, students with higher levels of classroom structure and peer support tend to be actively engaged and seek peer support, and these students have lower academic burnout [112].

### 4.3. Limitations

It needs to be acknowledged that there are still some limitations to this study. First, the sample of this study was mainly in liberal arts majors, with a large number of female students, and the inadequacy of the sample may affect the robustness of the study. Second, although the Classroom Burnout Inventory developed in this study was adapted to classroom teaching and has good reliability and validity, it still has room for further optimization. Third, as a self-report questionnaire, students’ answers may be influenced by interpersonal relationships, social expectations, and subjective interpretations of Likert-type answers [113]. Fourth, the impact of digital technology on college students’ psychological burnout at different stages of development may show temporal variability. The current data were collected from a single point in time, and the cross-sectional study cannot draw causal relationships from statistical relationships; therefore, a longitudinal design is needed in the future to elucidate the relationships between the variables under study.

Despite these limitations, the current study shows that technological developments do pose risks and challenges to young people’s mental health in the learning process. It is worthwhile to reflect on the side effects of digital technology overreach on traditional education.

## 5. Conclusions

Digital technology is being integrated deeply into China’s university education, and this process has been somewhat accelerated by the COVID-19 pandemic. After two years of intermittent home quarantine and online learning, most Chinese university students were allowed to return to campus. However, digital teaching styles were retained and combined with traditional classroom teaching in an “online + offline” model. The excessive, inappropriate, and poorly integrated digital teaching and learning environment exposes university students to more serious mental health risks, including academic burnout, which is often overlooked.

This study focuses on students’ classroom learning scenarios in an attempt to explore the impact of digital education on university students’ academic burnout. The “Classroom Burnout Inventory” and “Classroom Burnout Causes Inventory” were designed to test the level of burnout and the factors influencing it, respectively. A total of 206 valid questionnaires were collected and the validity and reliability of the questionnaires were confirmed by *t*-tests. Through correlation and regression analyses, we attempted to identify the factors and pathways of academic burnout.

The findings of the study confirmed most of the hypotheses posed at the beginning of this article. First, the level of burnout among post-epidemic university students is moderately high, with an average score of 42.413 (16–80). The most prominent symptom is “emotional exhaustion”. Second, the level of academic burnout was not significantly correlated with the gender and grade of university students, and no obvious differences were found in the level of academic burnout among different major categories (arts or science) (Proved H1). Third, some of the questions in the inventory reported that college students expressed dissatisfaction with the excessive and inappropriate use of digital technologies in the classroom, and data related to digital teaching showed a significant association with students’ academic burnout (Proved H3). Fourth, college students’ academic burnout in the digital classroom was more correlated with the personal causes than with external causes such as teachers, schools, and the environment. Of the three manifestations of academic burnout, personal causes of classroom “misbehaviour” were particularly prominent, while external factors were more likely to cause students to feel physically and mentally exhausted—a slight departure from the initial hypothesis (Partially proved H2).

Overall, classroom digital teaching and learning showed a positive association with academic burnout among Chinese university students. The experience from China has implications for other countries as well, reminding policy makers and educational practitioners to pay attention to the mental and physiological health of university students who spend long periods of time on digital learning in the post-COVID-19 era. Technology cannot be blindly superimposed, nor can it be over-relied upon. University education should be more than just a “human-computer interaction”, but also a return to human connection and spiritual transmission—the essence of education.

## Figures and Tables

**Figure 1 ijerph-19-13403-f001:**
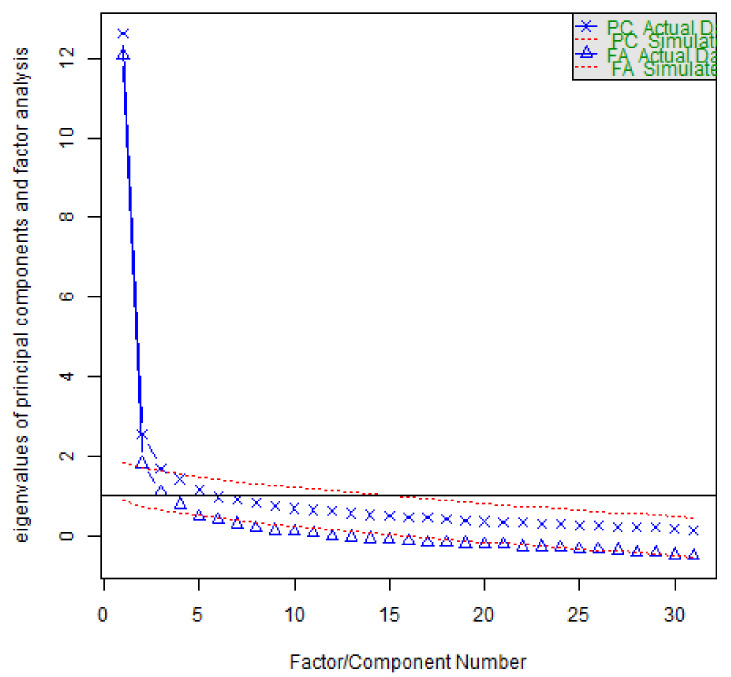
Parallel analysis of the questionnaire gravel plot.

**Figure 2 ijerph-19-13403-f002:**
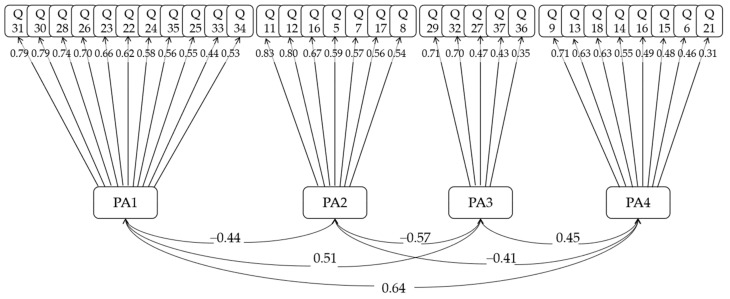
Factor analysis of the questionnaire.

**Figure 3 ijerph-19-13403-f003:**
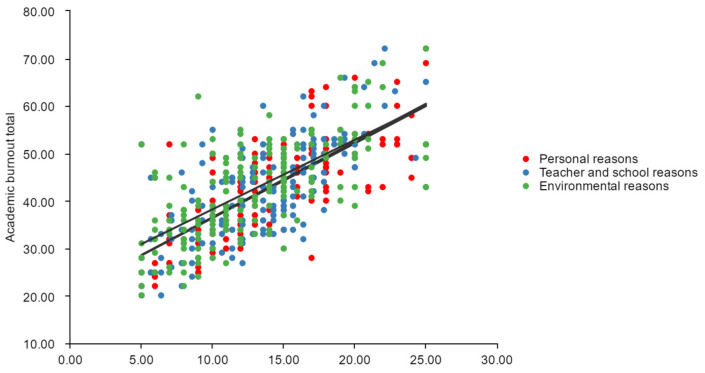
Relationship between academic burnout and personal, school, and environmental causes.

**Figure 4 ijerph-19-13403-f004:**
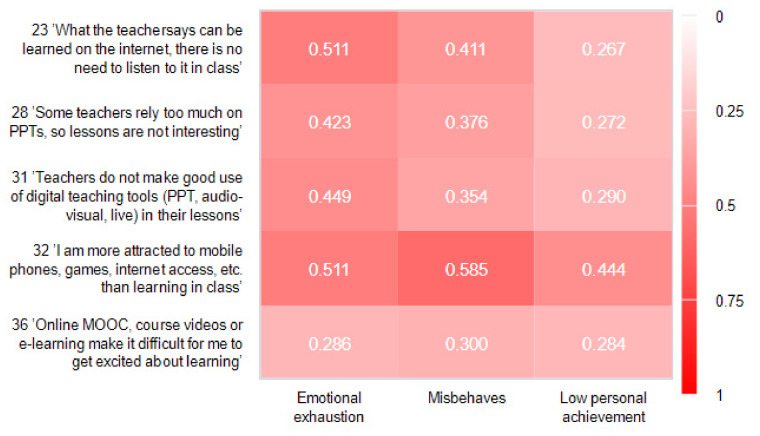
Pearson Correlation Analysis of Digital Instruction and College Students’ Classroom Burnout.

**Figure 5 ijerph-19-13403-f005:**
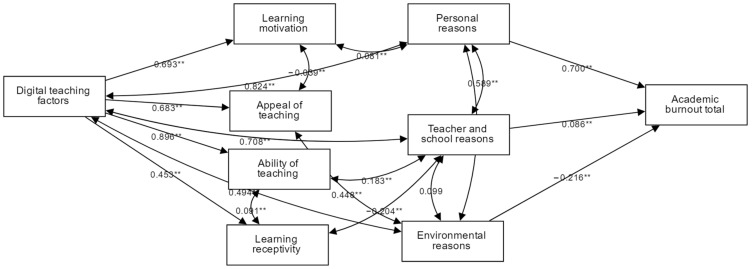
Model regression path (** *p* < 0.01).

**Table 1 ijerph-19-13403-t001:** Basic statistics of participants.

Items	Options	Frequency	Percentage (%)
(1) Gender	Male	62	30.10
	Female	144	69.90
(2) Grade (length of college admission)	0–1 year	29	14.08
	2 years	44	21.36
	3 years	54	26.21
	4 years	38	18.45
	4 years and above	41	19.90
(3) High school background	Partial liberal arts	117	56.80
	Partial science	66	32.04
	No distinction	23	11.17
(4) University major category	Liberal arts	176	85.44
	STEM (science)	30	14.56
Total		206	100.0

**Table 2 ijerph-19-13403-t002:** Basic statistics of each subscale.

Items	N of Samples	Min	Max	Mean	Std. Deviation	Median
Emotional Exhaustion	206	5.000	25.000	14.024	4.468	14.000
Misbehaves	206	5.000	25.000	13.010	3.722	13.000
Low Personal Achievement	206	5.000	25.000	13.655	4.086	13.571
Academic Burnout Total	206	20.000	72.000	42.413	9.748	43.000
Personal Causes	206	5.000	25.000	13.825	4.704	14.000
Teacher and School Causes	206	5.000	25.000	13.658	4.296	14.286
Environmental Causes	206	5.000	25.000	12.850	4.613	12.000

**Table 3 ijerph-19-13403-t003:** Parameter Estimates linear regression of the relationship between the three factors and academic burnout.

	Unstandardized Coefficients		Standardized Coefficients	t	*p*	VIF	R^2^	Adj R^2^	F
	B	Std. Error	Beta						
Constant	18.264	1.452	-	12.576	0.000 **	-	0.614	0.608	F (3, 202) = 107.019, *p* = 0.000
Personal causes	0.995	0.158	0.480	6.308	0.000 **	3.029			
Teacher and school causes	0.391	0.188	0.172	2.077	0.039 *	3.605			
Environmental causes	0.393	0.167	0.186	2.351	0.020 *	3.277			

Dependent Variable: Academic burnout total. D-W: 2.057; *n* = 206; * *p* < 0.05, ** *p* < 0.01.

**Table 4 ijerph-19-13403-t004:** Pearson Correlation analysis of the three types of burnout with the three causes.

		Emotional Exhaustion	Misbehaves	Low Personal Achievement
Personal causes	Coefficient	0.678 **	0.719 **	0.558 **
	*p* value	0.000	0.000	0.000
Teacher and school causes	Coefficient	0.655 **	0.598 **	0.484 **
	*p* value	0.000	0.000	0.000
Environmental causes	Coefficient	0.653 **	0.566 **	0.437 **
	*p* value	0.000	0.000	0.000

** *p* < 0.01.

## Data Availability

Data is not publicly available due to anonymity concerns. Readers interested in the data can contact the first or the corresponding author upon reasonable request.

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
