# Peer review of "Classroom Digital Teaching and College Students’ Academic Burnout in the Post COVID-19 Era: A Cross-Sectional Study"

_ijerph, 2022, doi:10.3390/ijerph192013403_

Round 1

Reviewer 1 Report

The study presented here provides interesting insights into the lessons learned in the pandemic about academic learning processes and the use of digital tools. However, key findings of the study regarding the influence of the quality of teaching, the learning environment and the personal prerequisites of the students on the learning success and the motivation of the learners also apply to non-digital teaching and learning.

When talking about "contemporary learning theory", it should be specified which theory is meant. There is not ONE "contemporary learning theory".

Furthermore, the references to Deci & Ryan are missing in the mentions of "self-determination theory.

A better structured and clearer reference to the hypotheses mentioned at the beginning would be desirable in the conclusion.

Author Response

Revision List

Dear Reviewer,

Thank you very much for your professional and valuable advice! Here are the changes I have made based on your review comments.

The study presented here provides interesting insights into the lessons learned in the pandemic about academic learning processes and the use of digital tools. However, key findings of the study regarding the influence of the quality of teaching, the learning environment and the personal prerequisites of the students on the learning success and the motivation of the learners also apply to non-digital teaching and learning.

My Response:Thank you so much for your review and recommendation of our research! Your ideas also bring inspiration and point the way to the next step of our research.

When talking about "contemporary learning theory", it should be specified which theory is meant. There is not ONE "contemporary learning theory".

My Response:Thank you for pointing this out. The statement was indeed too general.

My revision:I have added specific names of relevant theories.(e.g. social shared cognition, contextual learning, activity theory, ecopsychology, case-based reasoning) (Lines 623-624)

Furthermore, the references to Deci & Ryan are missing in the mentions of "self-determination theory.

My Response: Thanks for the reminder! Deci & Ryan, as important contributors to the theory, should not be ignored.

My revision:I queried and read Deci & Ryan's articles and books, which cover a dozen articles from 1980-2021. Among them, there is a special book on "self-determination theory", which I included in my reference list.(line 172)

A better structured and clearer reference to the hypotheses mentioned at the beginning would be desirable in the conclusion.

My Response:I agree with your idea.The original conclusion is indeed too short and many sentences are repetitions of the summary. It needs to be rewritten.

My revision: Following your advice, I have rewritten the conclusion. To echo the three hypotheses at the beginning, I have given the results of the proof for each hypothesis in the conclusion.

Also, I have summarized the core ideas of the rest of the text in the conclusion, and I have described the research methodology and process.

The research theme and the research value of the article are also emphasized.(lines 703-742)

Thanks again to the reviewers for their suggestions! The article revised according to your suggestions is much better in terms of details and conclusions that are close to the journal's publication standards.

Reviewer 2 Report

1. The "digital teaching" in the title of the paper is too broad and too large.

2. The research significance and value of the paper are not highlighted.

3. The Part of Introduction is too long and the thesis is not focused. It is suggested that the key research issues to be solved, research value and significance should be clarified in the introduction, and then the content of literature review should be separated into a separate part. In addition, literature review should be written in accordance with the following logic:1) What is digital teaching? What is the status of its development? 2) What is academic burnout? How does academic burnout affect students? 3) Why does academic burnout occur? Does digital teaching affect academic burnout?

4. These research hypotheses are not quite consistent with the research question.

5.Some expressions are repetitive and irregular,such as panic, anxiety, and anxiety(L136),new pneumonia epidemic(L138 ).

......

Author Response

Revision List

Dear Reviewer,

Thank you very much for your professional and valuable advice! Here are the changes I have made based on your review comments.

  1. The "digital teaching" in the title of the paper is too broad and too large.

My Response:Your advice has inspired me. The term “digital teaching” is very broad and seems to make it difficult to focus on specific issues.

My revision: One option we came up with regarding this issue was to add a qualifier to the title to narrow it down. Change "digital teaching" to "digital classroom teaching". However, this option was opposed by other team members. They thought that "digital classroom teaching" sounded strange and inappropriate. Also, "digital teaching" has been used in other similar articles.

After discussion, we made up our mind to keep the original title, but try to be more specific in the abstract, key sections and conclusion. This would limit the discussion of digital education technology to the specific context of the school classroom.The reader can better grasp the meaning of "digital teaching" in this article by combining the contents discussed in the text.

And you can see our efforts in the summary and conclusion sections of the article. (Lines 15-17, 727-729)

  1. The research significance and value of the paper are not highlighted.

My Response:Your reminder is very valuable! Since my introduction was too long, I neglected to emphasize the significance of the research.

My revision: I have restructured the introduction. At the beginning I emphasized the seriousness of academic burnout among college students in the post-epidemic period. It also directs the attention to digital teaching and learning, which is commonly overlooked. Rethinking digital approaches to education can help us better maintain the mental health and academic performance of college students. (Lines 31-37,43-46, 72-74)

In the conclusion section, I re-emphasize the research value of the article and the implications of the Chinese experience. It call on policy makers and educational practitioners to pay more attention to the digital health of college students.(lines 736-742)

  1. The Part of Introduction is too long and the thesis is not focused. It is suggested that the key research issues to be solved, research value and significance should be clarified in the introduction, and then the content of literature review should be separated into a separate part. In addition, literature review should be written in accordance with the following logic:1) What is digital teaching? What is the status of its development? 2) What is academic burnout? How does academic burnout affect students? 3) Why does academic burnout occur? Does digital teaching affect academic burnout?

My Response: Thank you for your constructive comments! I'm also thinking about how I can make each part of the discussion more focused. The introduction contains too much content, such as the significance of the study, background of the study, literature review and hypothesis, etc. They should be better organized.

My revision:Following the ideas you provided, I have made the following adjustments.

(1) I split the original introduction and separated the literature review from it. Also the research significance of the article is highlighted at the beginning. (Line 87)

(2) I reorganized the order of the literature review into "2.1 Digital Teaching", "2.2 Academic Burnout", and 2.3 Relationship between the two. Each of these sections is organized according to the logical ideas you provided.

In the "2.1 Digital Teaching" section, the concept of "digital teaching" and its core areas of research are added. (Lines 88-192)

In "2.2 Academic Burnout", I added a definition of "academic burnout" and how it affects students (reduces motivation and affects grades). (Lines 154-158)

In "2.3 Digital Education and Academic Burnout," I focus on the scholarly perspectives on "technology," "teachers," "systems "and "environment" on the digital learning of Chinese university students. This section also provides an introduction that shows the context for the article's research. (Lines 192-293)

  1. These research hypotheses are not quite consistent with the research question.

My Response: I also felt a subtle sense of dissatisfaction. It seems that the research hypothesis only comes into the theme at H3.

The article wanted to find a relationship between the variables 'digital teaching' and 'academic burnout'.

The logic between the three hypotheses is that H1 is to exclude the influence of students' internal factors on burnout, H2 is to narrow down the causes of burnout to external factors. And H3 really focuses the variables on the impact of digital teaching and learning on burnout.In order to give a fuller picture of the information contained in the data, we have split the hypothesis into several parts. In fact, the ultimate focus is on H3.

My revision:

(1) To make H2 more relevant, I have added "digital techs".( lines 82-83)

(2) In the conclusion section, I strengthen the connection between the research hypotheses and the research questions. The three hypotheses each prove the core question from a different side. The research theme of the article is reunited.(lines 719-734)

5.Some expressions are repetitive and irregular,such as panic, anxiety, and anxiety(L136),new pneumonia epidemic(L138 ).

My Response: Thank you for pointing out these issues so carefully!

My revision:I have corrected the problems you pointed out. (line 35)

I have also identified other areas of detail and corrected them one by one. For example, the full names of abbreviations such as PTSD and CNNIC have been added(line 146 & 577), inconsistent tenses have been unified, and the doi codes of the literature have been completed(lines 785, 818, 835, 838, 874, 885). I have also added data from this study to the 'Discussion' section as a basis( lines 651 & 674).

Thanks again to the reviewers for their suggestions! The article revised according to your suggestions is much better in terms of details and structures that are close to the journal's publication standards.

Reviewer 3 Report

The research model is based exclusively on quantitative research, guided in-depth interviews with selected respondents could lead to answers to the questions of why personnel factors are so relevant. The relationships between the individually verified hypotheses could have been more precisely described. Technically, the verified hypotheses are treated correctly, the material has the potential for deeper analysis. But even in this form it is an interesting work.

I recommend finalizing the conclusion, it is too brief and general, there should be clearly formulated and argued methodological procedures, partial results from the tested hypotheses and a synthesis based on them. In addition, the conclusion part matches (repeats) the abstract, which I consider unacceptable.

Author Response

Revision List

Dear Reviewer,

Thank you very much for your professional and valuable advice! Here are the changes I have made based on your review comments.

The research model is based exclusively on quantitative research, guided in-depth interviews with selected respondents could lead to answers to the questions of why personnel factors are so relevant. The relationships between the individually verified hypotheses could have been more precisely described. Technically, the verified hypotheses are treated correctly, the material has the potential for deeper analysis. But even in this form it is an interesting work.

I recommend finalizing the conclusion, it is too brief and general, there should be clearly formulated and argued methodological procedures, partial results from the tested hypotheses and a synthesis based on them. In addition, the conclusion part matches (repeats) the abstract, which I consider unacceptable.

My Response

Thank you for your professional comments and constructive suggestions!

(1) The in-depth interview method you mentioned is indeed very applicable to the research. But two factors constrained me to abandon this project temporarily.

Firstly, during my research, the epidemic in China suddenly got out of hand and some of our students returned to home quarantine. Teachers were also unable to see each and every one of their students. The town where I live was also closed off for a time. It is only now that the situation has improved. Uncertain objective circumstances prevented me from conducting face-to-face in-depth interviews. And online interviews may not be as effective.

Secondly, I am a university teacher and my identity role prevents students from expressing themselves authentically. Perhaps in-depth interviews would instead lead to the opposite conclusion. When I tried to discuss the topic with the students, they seemed apprehensive and the conversation did not go well. I therefore preferred to use a large, anonymous questionnaire to collect data.

Perhaps in future, by selecting more appropriate interviewees and developing a more sensible interview outline, the results would have been different.

(2) It has to be admitted that our analysis of the data, while comprehensive, does not go deep enough. This is limited by three factors:

Firstly, by the limitations of our analysis tool (SPSS).

Secondly, and frankly, because of our team's lack of ability to dig deeper into the data and analyse it.

Thirdly, the 37 questions in the two questionnaires have a certain amount of repetition and the data itself is not rich enough, thus lacking the potential to dig deeper.

(3) With regard to the "precise description of eachhypothesis" you suggest, and the "rewriting of the conclusion", I will present the process of my revision for you.

My revision:

(1) I have rewritten the conclusion.

The link between the research question and the research hypothesis was strengthened.

To echo the three hypotheses at the beginning, I have given the results of the proof for each hypothesis in the conclusion.

Also, I have summarized the core ideas of the rest of the text in the conclusion, and I have described the research methodology and process.

The research theme and the research value of the article are also emphasized. Discussed the relevance of China's experience on digital teaching in post-epidemic era to other countries.(lines 703-742)

(2) I have cut down the long and unfocused introduction.

I split the original introduction and separated the literature review from it. Also the research significance of the article is highlighted at the beginning. (Line 87)

I reorganized the order of the literature review into "2.1 Digital Teaching", "2.2 Academic Burnout", and 2.3 Relationship between the two. Each of these sections is organized according to the logical ideas you provided.

In the "2.1 Digital Teaching" section, the concept of "digital teaching" and its core areas of research are added. (Lines 88-192)

In "2.2 Academic Burnout", I added a definition of "academic burnout" and how it affects students (reduces motivation and affects grades). (Lines 154-158)

In "2.3 Digital Education and Academic Burnout," I focus on the scholarly perspectives on "technology," "teachers," "systems "and "environment" on the digital learning of Chinese university students. This section also provides an introduction that shows the context for the article's research. (Lines 192-293)

(3) Some details have also been amended.

For example, the full names of abbreviations such as PTSD and CNNIC have been added(line 146 & 577), inconsistent tenses have been unified, and the doi codes of the literature have been completed(lines 785, 818, 835, 838, 874, 885). I have also added data from this study to the 'Discussion' section as a basis( lines 651 & 674).

Thank you again for your professional and insightful advice! Although I couldn't find a perfect solution to some tricky problems for the time being. I have followed your advice and tried my best to make this article more in line with the journal's publication standards and criteria.

Reviewer 4 Report

The research presented in this study is about college students' academic burnout in the post COVID-19 era. It is a very interesting research work oriented to shed light in wellbeing topics, once students are in a post-COVID context.

Comments and suggestions

Minor issues: there are some typos, e.g. …

Line 31.- ‘…on the Chinese Internet[1],’; an extra space is required after `’Internet’. The same applies to the other references in the paper.

Line 34 to 36.- perhaps too ‘and’ in the same sentence?

Line 90.- It is not written in text what the acronym CNNIC stands for (China Internet Network Information Center).

Line 125 or line 592 (‘Covid-19’); versus line 128, line 129 or line 548 (COVID-19). It should be homogenized in the text.

Line 187.- into ' cell phone dependence,' where;    into 'cell phone dependence', where

Line 240.- by ' falsely busy'. The;  by 'falsely busy'. The

Line 295.- an ‘and’ should be deleted. Perhaps the first one could be deleted?; another option is to delete ‘,’…

Line 302.- Inventory(CBCI),; Inventory (CBCI),

Line 397.- Perhaps including a line between the items of Table 1 would make it clearer, since the items have different numbers of options.

Line 520.- Figure 5.By conducting linear; Figure 5. By conducting linear

Line 539.- It is not written in text what the acronym PTSD stands for (Post-traumatic stress disorder).

Line 596.- ‘manage. in February 2022’; ‘manage. In February 2022’.

Line 731.- ‘…Halls J.Psychological stress...’. an extra space is required before ‘Psychological’

Some considerations …

Aim of the study

Line 2.- Digital Teaching and College Students' Academic Burnout …

Line 11.- This study focused on the relationship between the use of digital technology in classroom scenarios and students' academic burnout

Line 253.- The aim is to explore whether there is a relationship between digital teaching and academic burnout among college students.

The definition included in the abstract is slightly different, since expresses and idea (‘digital technology in classroom scenarios’) that may go beyond than just ‘teaching’.

Line 57 to line 97

When analyzing the 3 stages there is a mix of using present verbal tense and past verbal tense (and even future verbal tense in line 61), which makes the comprehension a little bit difficult.

Line 79 to line 97

This paragraph develops the idea that Smart Classroom was launched in China (3rt stage; year 2020), as far as I understand. Perhaps should be interesting to make a reference to the idea that the ‘smart classroom’ concept, was developed and implemented previously in other countries several years ago, besides including the potential benefits that smarts classrooms may offer.

Figure 1 (line 336) is included in the article, despite it is not mentioned in the text.

Figure 2 (line 338 and 339). Consider moving Figure 2 after line 346, once the figure have been cited in text of the article (in line 346).

Table 1 & Table 2 are included in the article, despite both tables are not mentioned in the text.

Participants and limitations

Most of the participants in the research were enrolled in ‘liberal art majors’, and also the background of the surveyed students is biased to ‘liberal arts’ and ‘liberal arts nor science’ (Table 1). In fact, the authors identify this limitation in line 639.

Despite not being representative (in terms of statistics), have you analyzed as a cluster the group of students oriented to ‘science’? Their perceptions match with students oriented to ‘art’? Just in the case that science profiles differed a lot from the art profiles, perhaps would be consistent to consider the science profiles as outliers, and present the results of your research work in a context of ‘art students’…

Discussion

It would be nice to include additional data collected through the survey in this section to enrich this article. Most of the ideas and statements included in this section are referring to contributions and data of other research studies.

References

Some DOI are missed, e.g.

13. The relationship between Internet Use Disorder, depression and burnout among Chinese and German college students. Addictive Behaviors. Volume 89, February 2019, Pages 188-199.

https://doi.org/10.1016/j.addbeh.2018.08.011

17. Everything in Moderation: Moderate Use of Screens Unassociated with Child Behavior Problems

Christopher J. Ferguson. December 2017Psychiatric Quarterly 88(4)

DOI:10.1007/s11126-016-9486-3

22. The Dark Side of Internet Use: Two Longitudinal Studies of Excessive Internet Use, Depressive Symptoms, School Burnout and Engagement Among Finnish Early and Late Adolescents

Katariina Salmela-Aro, Katja Upadyaya, Kai Hakkarainen, Kirsti Lonka, Kimmo Alho. 2017 Feb;46(2):343-357

DOI: 10.1007/s10964-016-0494-2

23. Psychological stress reactivity and future health and disease outcomes: A systematic review of prospective evidence

Turner, A.I., Smyth, N., Hall, S.J., Torres, S.J., Hussein, M., Jayasinghe, S.U., Ball, K. and Clow, A. 2020. Psychoneuroendocrinology. 114 104599.

https://doi.org/10.1016/j.psyneuen.2020.104599

27. Applying the demand–control–support model on burnout in students: A meta-analysis

Soyeon Kim, Hankyul Kim, Eun Hye Park, Boram Kim, Sang Min Lee, Boyoung Kim First published: 17 August 2021

https://doi.org/10.1002/pits.22581

...

Author Response

Revision List

Dear Reviewer,

Thank you very much for your professional and valuable advice! Here are the changes I have made based on your review comments.

The research presented in this study is about college students' academic burnout in the post COVID-19 era. It is a very interesting research work oriented to shed light in wellbeing topics, once students are in a post-COVID context.

My Response: Thank you for your interest in my research and many sincere thanks for your suggestions for revision.

Comments and suggestions

Minor issues: there are some typos, e.g. …

Line 31.- ‘…on the Chinese Internet[1],’; an extra space is required after `’Internet’. The same applies to the other references in the paper.

My revision: Revised. And the remaining 87 similar issues in the article have been corrected.

Line 34 to 36.- perhaps too ‘and’ in the same sentence?

My revision: I used too many long parallel sentences, resulting in a lot of confusing 'and' in the text. I have fixed the problem you pointed out. A dozen other similar 'and's have been amended to use other expressions.

Line 90.- It is not written in text what the acronym CNNIC stands for (China Internet Network Information Center).

My revision: Full name has been added as your suggestion

Line 125 or line 592 (‘Covid-19’); versus line 128, line 129 or line 548 (COVID-19). It should be homogenized in the text.

Line 187.- into ' cell phone dependence,' where;    into 'cell phone dependence', where

Line 240.- by ' falsely busy'. The;  by 'falsely busy'. The

My revision: Similar misuse of punctuation and spaces has been corrected, both in the text and in the references.

Line 295.- an ‘and’ should be deleted. Perhaps the first one could be deleted?; another option is to delete ‘,’…

My revision: Delete "," and replace the second "and" with "as well as"(line 319)

Line 302.- Inventory(CBCI),; Inventory (CBCI),

Line 397.- Perhaps including a line between the items of Table 1 would make it clearer, since the items have different numbers of options.

My revision: I have to say you are a really professional and careful reviewer! The table has become much clearer after the changes you suggested.

Line 520.- Figure 5.By conducting linear; Figure 5. By conducting linear

My revision: Corrected as you suggested. And corrected other similar errors.

Line 539.- It is not written in text what the acronym PTSD stands for (Post-traumatic stress disorder).

My revision: Full name has been added as your suggestion

Line 596.- ‘manage. in February 2022’; ‘manage. In February 2022’.

Line 731.- ‘…Halls J.Psychological stress...’. an extra space is required before ‘Psychological’

My revision: Thank you for your careful advice, I have corrected them.

Some considerations …

Aim of the study

Line 2.- Digital Teaching and College Students' Academic Burnout …

Line 11.- This study focused on the relationship between the use of digital technology in classroom scenarios and students' academic burnout

Line 253.- The aim is to explore whether there is a relationship between digital teaching and academic burnout among college students.

The definition included in the abstract is slightly different, since expresses and idea (‘digital technology in classroom scenarios’) that may go beyond than just ‘teaching’.

My revision: Yes, as you say more, the inconsistencies in presentation can be confusing. I have changed them all to "digital teaching" for consistency. (line 12)

Line 57 to line 97

When analyzing the 3 stages there is a mix of using present verbal tense and past verbal tense (and even future verbal tense in line 61), which makes the comprehension a little bit difficult.

My revision: Thank you for pointing out my grammatical errors. I have unified them into the past tense. (lines 138-145)

Line 79 to line 97  This paragraph develops the idea that Smart Classroom was launched in China (3rt stage; year 2020), as far as I understand. Perhaps should be interesting to make a reference to the idea that the ‘smart classroom’ concept, was developed and implemented previously in other countries several years ago, besides including the potential benefits that smarts classrooms may offer.

My revision: As per your suggestion, I have added the introduction of the Smart Classroom. This will help the reader to understand it better.(lines 135-140)

Figure 1 (line 336) is included in the article, despite it is not mentioned in the text.

Figure 2 (line 338 and 339). Consider moving Figure 2 after line 346, once the figure have been cited in text of the article (in line 346).

Table 1 & Table 2 are included in the article, despite both tables are not mentioned in the text.

My revision: Thank you for your suggestions, which have helped make this article more reader-friendly. I have adjusted the table and the figures where it is first mentioned. And all images and tables are mentioned in the text. (lines 367, 414, 443)

Participants and limitations

Most of the participants in the research were enrolled in ‘liberal art majors’, and also the background of the surveyed students is biased to ‘liberal arts’ and ‘liberal arts nor science’ (Table 1). In fact, the authors identify this limitation in line 639.

Despite not being representative (in terms of statistics), have you analyzed as a cluster the group of students oriented to ‘science’? Their perceptions match with students oriented to ‘art’? Just in the case that science profiles differed a lot from the art profiles, perhaps would be consistent to consider the science profiles as outliers, and present the results of your research work in a context of ‘art students’…

My revision: Before starting the survey, we also expected that students in different major categories (arts or science) would show variability in burnout. However, probably due to the insufficient sample size or the lack of differentiation in the questionnaire design, we did not find significant differences in the data.

As many of you may also be interested in this issue, I have added data and analysis on it ( lines 459-461). I also mentioned this finding again in the conclusion section (lines 722-725).

Discussion

It would be nice to include additional data collected through the survey in this section to enrich this article. Most of the ideas and statements included in this section are referring to contributions and data of other research studies.

My revision: Your advice is very valuable. I have added some data from this survey to the Discussion as the basis. (lines 589-590, 650-652, 673-674)

References

Some DOI are missed, e.g.

  1. The relationship between Internet Use Disorder, depression and burnout among Chinese and German college students. Addictive Behaviors. Volume 89, February 2019, Pages 188-199.

https://doi.org/10.1016/j.addbeh.2018.08.011

  1. Everything in Moderation: Moderate Use of Screens Unassociated with Child Behavior Problems

Christopher J. Ferguson. December 2017Psychiatric Quarterly 88(4)

DOI:10.1007/s11126-016-9486-3

  1. The Dark Side of Internet Use: Two Longitudinal Studies of Excessive Internet Use, Depressive Symptoms, School Burnout and Engagement Among Finnish Early and Late Adolescents

Katariina Salmela-Aro, Katja Upadyaya, Kai Hakkarainen, Kirsti Lonka, Kimmo Alho. 2017 Feb;46(2):343-357

DOI: 10.1007/s10964-016-0494-2

  1. Psychological stress reactivity and future health and disease outcomes: A systematic review of prospective evidence

Turner, A.I., Smyth, N., Hall, S.J., Torres, S.J., Hussein, M., Jayasinghe, S.U., Ball, K. and Clow, A. 2020. Psychoneuroendocrinology. 114 104599.

https://doi.org/10.1016/j.psyneuen.2020.104599

  1. Applying the demand–control–support model on burnout in students: A meta-analysis

Soyeon Kim, Hankyul Kim, Eun Hye Park, Boram Kim, Sang Min Lee, Boyoung Kim First published: 17 August 2021

https://doi.org/10.1002/pits.22581

My revision: In addition to what you mentioned, I rechecked and added 21 missing doi codes. (lines 757-984)

I am very grateful. You are one of the best reviewers I have ever had. You were professional and careful, pointing out almost all the detail errors, which took my article a big step closer to publication standards.

Every aspect of my article has been improved by your suggestions, thank you very much!

Round 2

Reviewer 2 Report

1)The logic between each paragraph in the Introduction is not strong, and there is no clear statement highlighting the research status, research problem, research value and significance, etc.

2) The Digital Teaching in the title of the paper is too broad and too large. Moreover, there is no specific and clear definition of Digital Teaching in the Literature Review.

3)The innovation of the research is not strong, and the data analysis methods are relatively general

Author Response

The logic between each paragraph in the Introduction is not strong, and there is no clear statement highlighting the research status, research problem, research value and significance, etc.

My revision:Thank you for your professional and insightful suggestions. In response to your first round of comments, I have highlighted the research value of the article in the article (lines 31-37, 43-46, 72-74) and re-emphasised the relevance of the Chinese experience to digital health in the conclusion.

In this round of revisions, I have enhanced the logic of the introductory section and clearly identified the current state of research, research questions and research implications of the article.

(1) The paragraph on the current state of research was added. In the second paragraph (lines 38-51), I briefly summarised the current state of research on the issue, adding some literature. A logically incoherent paragraph in the original text has been removed.

(2) Highlighting the research question. In the fourth paragraph (lines 62-70) I introduce the question to be addressed in the paper and begin the fifth paragraph with a clear statement of the core research objectives of the article.‘The study focuses on undergraduate digital education in China in the post-COVID-19 period. The goal is to investigate if academic burnout among college students is associated with digital classroom teaching.’

(3) A paragraph on the significance of the study has been added. In the fifth paragraph (lines 71-79) the value and significance of the study is stated in three points. First, this study may be relevant in identifying the role of digital technology in college students' learning processes and investigating the real reasons behind their academic burnout. Second, an overview of the manner in which the weariness of university students is impacted by digital education is useful. Thirdly, it offers methods to prevent and treat mental health crises brought on by excessive classroom digital teaching and learning, such as management approaches, instructional tactics, and psychiatric therapy.

The Digital Teaching in the title of the paper is too broad and too large. Moreover, there is no specific and clear definition of Digital Teaching in the Literature Review.

My revision:Thank you for your suggestions.

(1) About the title.

The article focuses on the negative impact of the "online + offline" digital teaching model on students' psychology in the post-epidemic Chinese university classroom. We restrict the scenario of digital teaching to the "classroom" and revise the title to "Classroom Digital Teaching and College Students' Academic Burnout in the Post COVID-19 Era". In the title and conclusion of the literature review, all references to "digital teaching" were replaced with "classroom digital teaching".

This has three improvements: firstly, it makes the title of the article more specific, and secondly, the term 'classroom digital teaching' is consistent with the Classroom Burnout Inventory (CBI) questionnaire we designed. Thirdly, "digital teaching and learning in the classroom" is logically aligned with the discussion on improving the use of digital technology in the classroom in the "4. discussion”.

2) Definition and literature review of digital teaching and learning.

In section 2.1, I have added a definition of digital teaching and learning (lines 91-93).Digital teaching is the practice of teachers and students engaging in instructional activities inside a digital learning environment while adhering to contemporary educational theories and guidelines, employing digital teaching resources, and utilizing digital teaching techniques to develop talents.

Also added to this section is an extensive review of the literature on digital teaching and learning in the classroom, particularly the discussions and research findings in the post-epidemic era (lines 94-136).

The innovation of the research is not strong, and the data analysis methods are relatively general.

My response: It has to be admitted that our analysis of the data, while comprehensive, does not go deep enough. This is limited by three factors:

Firstly, by the limitations of our analysis tool (SPSS). Secondly, and frankly, because of our team's lack of ability to dig deeper into the data and analyse it. Thirdly, the 37 questions in the two questionnaires have a certain amount of repetition and the data itself is not rich enough, thus lacking the potential to dig deeper.

Your suggestions therefore point the way to the next steps in our research. We will design a more in-depth research programme in future studies and process the data with more precise analytical tools.

Thank you again for your professional and patient review!